# POSTRAINBENCH: A COMPREHENSIVE BENCHMARK AND A NEW MODEL FOR PRECIPITATION FORECASTING

## ABSTRACT

Accurate precipitation forecasting is a vital challenge of both scientific and societal importance. Data-driven approaches have emerged as a widely used solution for addressing this challenge. However, solely relying on data-driven approaches has limitations in modeling the underlying physics, making accurate predictions difficult. Coupling AI-based post-processing techniques with traditional Numerical Weather Prediction (NWP) methods offers a more effective solution for improving forecasting accuracy. Despite previous post-processing efforts, accurately predicting heavy rainfall remains challenging due to the imbalanced precipitation data across locations and complex relationships between multiple meteorological variables. To address these limitations, we introduce the **PostRainBench**, a comprehensive multi-variable NWP post-processing benchmark consisting of three datasets for NWP post-processing-based precipitation forecasting. We propose **CAMT**, a simple yet effective Channel Attention Enhanced Multi-task Learning framework with a specially designed weighted loss function. Its flexible design allows for easy plug-and-play integration with various backbones. Extensive experimental results on the proposed benchmark show that our method outperforms state-of-the-art methods by 6.3%, 4.7%, and 26.8% in rain CSI on the three datasets respectively. Most notably, our model is the first deep learning-based method to outperform traditional Numerical Weather Prediction (NWP) approaches in extreme precipitation conditions. It shows improvements of 15.6%, 17.4%, and 31.8% over NWP predictions in heavy rain CSI on respective datasets. These results highlight the potential impact of our model in reducing the severe consequences of extreme weather events.

## 1 INTRODUCTION

Precipitation forecasting (Sønderby et al., 2020; Espeholt et al., 2022) refers to the problem of providing a forecast of the rainfall intensity based on radar echo maps, rain gauge, and other observation data as well as the Numerical Weather Prediction (NWP) models (Shi et al., 2017). Accurate rainfall forecasts can guide people to make optimal decisions in production and life. The frequency and intensity of rainfall varies based on geography. Though the occurrence of extreme precipitation events is relatively infrequent, they can lead to adverse impacts on both agricultural production and community well-being (de Witt et al., 2021).

At present, the most accurate forecasting system is the Numerical Weather Prediction (NWP) method (Bi et al., 2023), which represents atmospheric states as discretized grids and numerically solves partial differential equations that describe the transition between those states. NWP predictions cover a wide range of variables. Each of them provides information about meteorological states (wind speed, temperature, pressure, etc.) and surface states (water vapor on the surface, snow amount, etc.), presenting a multidimensional description of the atmospheric state.

In the past few years, geoscience has begun to use deep learning to better exploit spatial and temporal structures in the data. Studies are beginning to apply combined convolutional–recurrent approaches to geoscientific problems such as precipitation nowcasting (Shi et al., 2015; Wang et al., 2017; Shi et al., 2017). Precipitation nowcasting is mainly based on data-driven extrapolation and lacks physics-based modeling (Kim et al., 2022). Despite continuous efforts to directly enhance global weather forecasting (Lam et al., 2022; Bi et al., 2023; Chen et al., 2023a;b) with deep learning

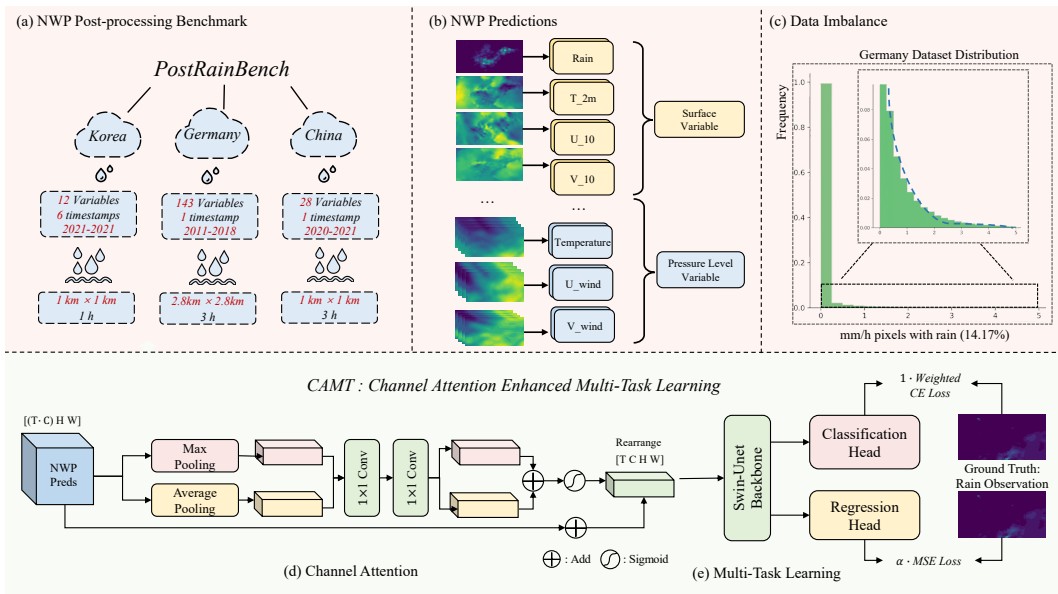

Figure 1: An overview of the proposed **PostRainBench** and **CAMT** framework. (a) illustrates our benchmark's attributes. (b) shows the input composition. (c) presents the distribution of the German dataset, highlighting the data imbalance challenge. The bottom section illustrates our CAMT workflow: (d) NWP inputs undergo processing by the Channel Attention Module, followed by a Swin-Unet backbone. (e) Multi-task learning with hybrid weighted loss using classification and regression heads.

methods, on the NWP side, post-processing methods can be developed to alleviate the predictable biases of NWP models. Combining AI-based and NWP methods can bring about both strengths for a stronger performance (Bi et al., 2023).

For the post-processing task, NWP predictions are fed to a deep learning model which is trained to output refined precipitation forecasts. Rainfall station observations are used as ground truth. In a nutshell, the overall goal is to post-process the predictions from NWP using deep models, under the supervision of rainfall station observations.

However, Post-NWP optimization poses several distinct challenges that distinguish it from typical weather forecasting optimization problems and computer vision tasks.

**Variable Selection and Modeling.** In NWP, each pixel on the grid has various variables expressing the atmospheric feature state, which exhibit different statistical properties. This discrepancy includes spatial dependence and interdependence among variables, which violate the crucial assumption of identical and independently distributed data (Reichstein et al., 2019). The variables exhibit high correlation among themselves and also possess a degree of noise. Previous approaches have either used all available variables (Rojas-Campos et al., 2022) as input or relied on expert-based variable selection (Kim et al., 2022), which did not fully leverage the modeling capabilities.

**Class Imbalance.** The distribution of precipitation exhibits a significant imbalance, making model optimization challenging. A prior study (Shi et al., 2017) introduced WMSE, which assigned higher weighting factors to minority classes. Another study (Cao et al., 2022b) combined a reweighting loss with the MSE loss to mitigate the degradation in performance for majority classes. While these approaches have succeeded in improving forecast indicators for the minority class (heavy rainfall), they have inadvertently compromised the model's performance on the majority class.

**Lack of A Unified Benchmark.** A previous study, KoMet (Kim et al., 2022), introduced a small dataset covering the time span of two years. Due to the limited data samples, models trained solely on such datasets may risk overfitting to specific data characteristics. Furthermore, KoMet only selected a subset of NWP variables as input. In contrast, another study (Rojas-Campos et al., 2022)

utilized all 143 available NWP variables as input. The limited size of the dataset, along with the lack of a standardized method for selecting variables, hinders research progress in improving the NWP post-processing task.

To tackle the aforementioned challenges, we introduce a new model learning framework and a unified benchmark for robust evaluation.

We summarize our contributions as follows:

- We introduce **PostRainBench**, a comprehensive multi-variable benchmark, which covers the full spectrum of scenarios with and without temporal information and various combinations of NWP input variables. This unified benchmark could help accelerate the research area of NWP post-processing-based precipitation forecasting.

- We propose **CAMT**, a simple yet effective Channel Attention Enhanced Multi-task Learning framework with a specially designed weighted loss function. CAMT is flexible and can be plugged into different model backbones.

- On the proposed benchmark, our model outperforms state-of-the-art methods by **6.3%**, **4.7%**, and **26.8%** in rain CSI on three datasets, respectively. Furthermore, it's worth highlighting a significant milestone achieved by our model. It stands as the first deep learning model to surpass NWP method in heavy rain, with improvements of **15.6%**, **17.4%**, and **31.8%** over NWP predictions across respective datasets. This underscores its potential to effectively mitigate substantial losses in the face of extreme weather events.

## 2 RELATED WORK

**Deep Learning-based Precipitation Nowcasting** Regarding precipitation nowcasting as a spatiotemporal sequence forecasting problem, Shi et al. (2015) first proposed Convolutional Long Short-Term Memory (ConvLSTM) to directly predict the future rainfall intensities based on the past radar echo maps. PredRNN (Wang et al., 2017) separated the spatial and temporal memory and communicated them at distinct LSTM levels. Lebedev et al. (2019) used the U-Net architecture (Ronneberger et al., 2015) to nowcast categorical radar images with results that outperformed traditional nowcasting methods. MetNet (Sønderby et al., 2020) employed a combination of a convolutional long short-term memory (LSTM) encoder and an axial attention decoder, which was demonstrated to achieve strong results on short-term low precipitation forecasting using radar data.

**NWP Post-processing** A recent work (Rojas-Campos et al., 2022) used Conditional Generative Adversarial Network (CGAN) (Goodfellow et al., 2014) to post-process NWP predictions to generate precipitation maps and compared it with U-Net (Ronneberger et al., 2015) and two deconvolution networks. Following the Critical Success Index(CSI) scores, there was an initial high performance in low precipitation forecasting and a progressive decline as the threshold increased, indicating a general difficulty in predicting high precipitation events. NWP's direct predictions of rain presented the highest scores in predicting high precipitation events over proposed deep learning methods. Another work (Kim et al., 2022) proposed an open dataset with selected NWP predictions as input and compared the performance of three baseline models, U-Net, ConvLSTM and MetNet. The findings were similar, while deep learning models achieved better performance in low rain forecasting, none of the deep learning models surpassed the performance of NWP in heavy rain conditions.

## 3 POSTRAINBENCH

### 3.1 TASK FORMULATION

In this study, we consider optimizing the following model:

$$\min_{\boldsymbol{w}} \left\{ \mathcal{L}(\boldsymbol{w}; \mathcal{D}) \triangleq \mathbb{E}_{(X_t, y_t) \sim \mathcal{D}}[\ell(y_t; F(X_t, \boldsymbol{w}))] \right\} \tag{1}$$

where $\mathcal{L}$ represents the objective function parameterized by $\boldsymbol{w}$ on the dataset $\mathcal{D}$. As shown in Figure 6, the input is NWP predictions $X_t$, the corresponding ground-truth is rain observation $y_t$ at time $t$, and $\ell$ denotes the loss function between the output of our proposed model $F(\cdot, \boldsymbol{w})$ and

the ground-truth. The NWP predictions $X_t$ are derived from the NWP model at time $t - L - \tau$, constituting a sequence denoted as $X_t = \boldsymbol{x}_{(t-L)}, \boldsymbol{x}_{(t-L+1)}, \cdots, \boldsymbol{x}_{(t-2)}, \boldsymbol{x}_{(t-1)}$, where $L$ signifies the sequence length and $\tau$ denotes the lead time. Our post-process model $F(\cdot, \boldsymbol{w})$ takes the sequence of NWP predictions $X_t$ as input, aiming to predict a refined output $\tilde{y}_t$ (at time $t$), where the rainfall observations $y_t$ (at time $t$) sever as ground truth to train our model. In our multi-task framework, the prediction of our model at time $t$ is defined as a classification forecast $\tilde{\boldsymbol{y}}_{cls}$ and a regression forecast $\tilde{\boldsymbol{y}}_{reg}$. Our proposed model $F(\cdot, \boldsymbol{w})$ is formulated as:

$$\tilde{\boldsymbol{y}}_{cls}, \tilde{\boldsymbol{y}}_{reg} = F(X_t, \boldsymbol{w}) \tag{2}$$

$$= F(\{\boldsymbol{x}_{(t-L)}, \boldsymbol{x}_{(t-L+1)}, \cdots, \boldsymbol{x}_{(t-2)}, \boldsymbol{x}_{(t-1)}\}, \boldsymbol{w}) \tag{3}$$

where $\boldsymbol{w}$ is the trainable parameters. Our model utilizes a classification head and a regression head to generate two final forecasts, $\tilde{\boldsymbol{y}}_{cls}$ and $\tilde{\boldsymbol{y}}_{reg}$. $\tilde{\boldsymbol{y}}_{cls}$ is a probability matrix and each item indicates the probability of a specific class among {'non-rain', 'rain', 'heavy rain'}. $\tilde{\boldsymbol{y}}_{reg}$ is a prediction value of each pixel in the grid.

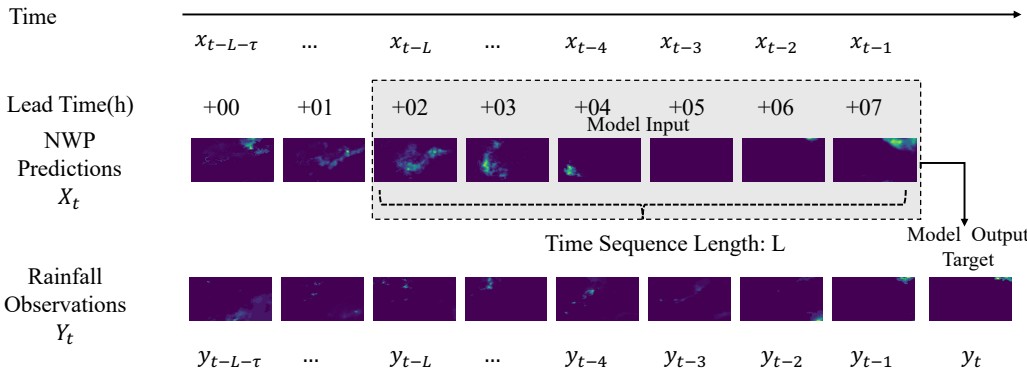

Figure 2: An illustrate of NWP post-processing task. NWP predictions $X_t$ with a time sequence length of $L$ is used as input, while rain observation $y_t$ is used as ground truth.

## 3.2 EVALUATION METRICS

In terms of evaluation, we adopt commonly used multi-class classification metrics for precipitation forecasting by previous works (Kim et al., 2022). The evaluation metrics are calculated based on the number of true positives ($TP_k$), false positives ($FP_k$), true negatives ($TN_k$), and false negatives ($FN_k$) for some generic class $k$. We describe the main metrics we consider as follows:

**Critical Success Index (CSI)** (Donaldson et al., 1975) is a categorical metric that takes into account various elements of the confusion matrix, similar with F1-score having the value as $\frac{TP_k}{TP_k + FN_k + FP_k}$.

**Heidke Skill Score (HSS)** (Woo & Wong, 2017) as stated by Hogan et al. (2010), is more equitable in evaluating the forecasting performance. Higher HSS means better performance and a positive HSS indicates that a forecast is better than a random-based forecast. HSS is calculated as $\frac{2 \times (TP_k \times TN_k - FN_k \times FP_k)}{FP_k^2 + TN_k^2 + 2 \times TP_k \times FN_k + (FP_k + TN_k)(TP_k + FP_k)}$.

Please refer to the Section A.4 for all metrics we use in the benchmark.

## 3.3 DATASETS

To address the issues of limited dataset size and the lack of a standardized criterion for variable selection, we introduce a unified benchmark comprising three datasets. Two of these datasets are sourced from prior research, while the third is collected from a public challenge. We describe our processing and standardization of the datasets below.

The first dataset, called KoMet (Kim et al., 2022), was collected in South Korea. The input data originates from GDAPS-KIM, a global numerical weather prediction model that furnishes hourly forecasts for diverse atmospheric variables. GDAPS-KIM operates at a spatial resolution of 12 km $\times$ 12 km, resulting in a spatial dimension of $65 \times 50$. The variables fall into two categories: pressure level variables and surface variables. For benchmarking purposes, 12 variables out of the 122 are selected according to Korean experts, and we follow this setting in our paper.

The second dataset originates from Germany (Rojas-Campos et al., 2022). This dataset covers the period from 2011 to 2018 and is confined to a selected area in West Germany. The input data is derived from the COSMO-DE-EPS forecast (Peralta et al., 2012), which provides 143 variables of the atmospheric state. For this dataset, the forecast with a 3-hour lead time is selected. A detailed description of the COSMO-DE-EPS output can be found in Schättler et al. (2008). The input data has a spatial resolution of $36 \times 36$, while the output data is available at a resolution of $72 \times 72$. To give a fair comparison between various algorithms, we perform interpolation on both to bring them to a consistent resolution of $64 \times 64$.

The third dataset originates from China and provides hourly, 1 km $\times$ 1 km resolution, 3-hour grid point precipitation data for the rainy season. This dataset spans from April to October in both 2020 and 2021. Additionally, it includes 3-hour lead time forecasts from a regional NWP model, with 28 surface and pressure level variables such as 2-meter temperature, 2-meter dew point temperature, 10-meter u and v wind components, and CAPE (Convective Available Potential Energy) values. For all variables provided, please refer to Table 6. Each time frame in this dataset covers a substantial spatial area, featuring a grid size of $430 \times 815$. To maintain consistency, we interpolate this dataset to a more manageable $64 \times 64$ grid.

We summarize important details of the three datasets in the Table 5.

## 3.4 DATA DISTRIBUTION

We analyze the distribution of the observed precipitation data, which serves as the ground truth, across the three datasets. In accordance with the framework outlined in Kim et al. (2022), we categorize precipitation into two types: rain and heavy rain, each with its set of evaluation metrics and frame this forecasting problem as a three-class classification task. It is important to note that the threshold for defining heavy rain can vary by location due to differences in rainfall frequency influenced by geographical and climatic factors.

Table 1: Statistics of three datasets.

| Dataset | Rain rate (mm/h) | Proportion (%) | Rainfall Level |
|---|---|---|---|
| KoMet | $[0.0, 0.1)$ | 87.24 | No Rain |
| | $[0.1, 10.0)$ | 11.57 | Rain |
| | $[10.0, \infty)$ | 1.19 | Heavy Rain |
| Germany | $[0.0, 10^{-5})$ | 85.10 | No Rain |
| | $[10^{-5}, 2.0)$ | 13.80 | Rain |
| | $[2.0, \infty)$ | 1.10 | Heavy Rain |
| China | $[0.0, 0.1)$ | 91.75 | No Rain |
| | $[0.1, 2.0)$ | 3.81 | Rain |
| | $[2.0, \infty)$ | 4.44 | Heavy Rain |

In Germany dataset, Rojas-Campos et al. (2022) explores various thresholds including 0.2, 0.5, 1, 2, and 5, we adopt a rain threshold of $10^{-5}$ mm/h since its distribution is concentrated in [0,1] and we adhere to the rain threshold of 0.1mm/h adopted by Kim et al. (2022) In Korea dataset, we adhere previous heavy rain threshold of 10mm/h and opt for a unified threshold of 2mm/h in another two datasets, enabling a more equitable comparison. The distribution and the rain categorization of the three datasets are presented in Table 1. It is evident that all three datasets exhibit significant imbalances, which presents a great challenge to predict extreme weather scenarios.

## 4 METHOD

As illustrated in Figure 1, our model can be divided into three parts. The first part is a channel attention module (Woo et al., 2018). The second part is the Swin-Unet backbone (Cao et al., 2022a) that generates linear projections. The third part is a multi-task learning branch with a hybrid loss. We describe the first and third parts in detail below and put explanations of Swin-Unet in Section A.1 and A.3.

### 4.1 CHANNEL ATTENTION MODULE

While the data-driven approaches can be improved by incorporating more variables, they also escalate the storage space and memory demands for modeling. Previous approaches either used all available variables as input or relied on expert-driven variable selection, which did not fully harness modeling capabilities.

A recent study (Chen et al., 2023a) viewed the medium-range forecast problem from a multi-modal perspective and used a cross-model transformer to fuse different modalities. It inspires us by emphasizing that the crucial aspect of accurate multi-variable weather forecasting lies in effectively modeling the relationships between different channels (variables).

To this concern, we introduce the Channel Attention Module (CAM), which enables variable selection for a unified NWP post-processing task, and models intricate relationships between variables.

CAM aggregates spatial information of a feature map by using both average-pooling and max-pooling operations, generating two different spatial context descriptors: $\mathbf{F^c_{avg}}$ and $\mathbf{F^c_{max}}$. Both descriptors are forwarded to a shared multi-layer perceptron (MLP) to produce a channel attention map $\mathbf{M_c} \in \mathbb{R}^{C \times 1 \times 1}$. To reduce parameter overhead, the hidden activation size is set to $\mathbb{R}^{C/r \times 1 \times 1}$, where $r$ is the reduction ratio. After the shared network, the two output feature vectors are merged with element-wise summation. We employ a residual connection (He et al., 2016) by adding the attention map to the original input, which serves as the input for the subsequent backbone stage.

In short, the channel attention is computed as:

$$\begin{aligned}
\mathbf{M_c}(\mathbf{F}) &= \sigma(MLP(AvgPool(\mathbf{F})) + MLP(MaxPool(\mathbf{F}))) \\
&= \sigma(\mathbf{W_1}(\mathbf{W_0}(\mathbf{F^c_{avg}})) + \mathbf{W_1}(\mathbf{W_0}(\mathbf{F^c_{max}}))),
\end{aligned} \tag{4}$$

where $\sigma$ denotes the sigmoid function, $\mathbf{W_0} \in \mathbb{R}^{C/r \times C}$, and $\mathbf{W_1} \in \mathbb{R}^{C \times C/r}$. We choose $r = 16$. Note that the MLP weights, $\mathbf{W_0}$ and $\mathbf{W_1}$, are shared for both inputs and the activation function is followed by $\mathbf{W_0}$. We choose GeLU activation function instead of ReLU.

The resulting feature maps are then input to the Swin-Unet backbone, as shown in Figure 1.

The backbone model is connected to a classification head and a regression head, which are learned under our proposed multitask learning framework as described in the next section.

### 4.2 MULTITASK LEARNING WITH HYBRID WEIGHTED LOSS

Prior research has traditionally approached precipitation forecasting as either a regression or classification problem. In practice, people care more about the rain level than specific rain intensity. Considering the realistic needs, a classification task is more appropriate. However, In our practice, we find regression task can enhance the learning for classification task. This approach streamlines model optimization, aligning with the metrics we utilize.

In this paper, we introduce a combination of Mean Squared Error (MSE) loss and weighted Cross-Entropy (CE) loss within a multi-task learning framework, incorporating two task outputs $\tilde{\mathbf{y}}_{cls}, \tilde{\mathbf{y}}_{reg}$ with a hyperparameter $\alpha$. Utilizing dedicated classification and regression heads encourages the backbone to focus on learning essential features for both tasks.

As previously mentioned, precipitation forecasting grapples with the challenge of highly imbalanced class distributions from a classification standpoint. To tackle this issue, we apply class weights $w_c$ based on the class distribution of each dataset. The full loss function $L_{hybrid}$ is defined as:

$$L_{cls} = \sum_{i=1}^{h} \sum_{j=1}^{w} (- \sum_{c=1}^{M} w_c y_t \log(\tilde{\boldsymbol{y}}_{cls})) \tag{5}$$

$$L_{reg} = \sum_{i=1}^{h} \sum_{j=1}^{w} (\tilde{\boldsymbol{y}}_{reg} - y_t)^2 \tag{6}$$

$$L_{hybrid} = L_{cls} + \alpha L_{reg} \tag{7}$$

In Equation 5, $h$ and $w$ refer to the spatial resolution, height and width, $c$ refer to the grid's pixel class, $M$ is the number of classes, and $y_t$ is the ground truth.

## 5 EXPERIMENTAL EVALUATION

### 5.1 IMPLEMENTATION DETAILS

We compare our proposed Swin-Unet-based CAMT framework with various strong baselines, including the NWP method, three deep learning models (ConvLSTM, UNet, MetNet). Swin-Unet (Ronneberger et al., 2015) is a Unet-like Transformer. The tokenized image patches are fed into the Swin Transformer-based (Liu et al., 2021) U-shaped Encoder-Decoder architecture with skip connections for local-global semantic feature learning.

The datasets are split into training, validation, and test sets following the configurations outlined in previous studies. For the China dataset, we randomly partition the data into a 6:2:2 ratio. To ensure consistency with prior studies, we select the model with the best CSI performance on the validation set and report its performance on the test set. Each model is run with three different random seeds for robust performance. We use the Adam optimizer for all models.

For the Korea dataset, baseline models are trained with a learning rate of 0.001 (as mentioned in Kim et al. (2022)), while Swin-Unet models are trained with a learning rate of 0.0001. Consistent with previous settings, a batch size of 1 is employed, and all models are trained for 50 epochs. We apply a weight of [1, 5, 30] for the CE Loss. We utilize a hyperparameter $\alpha$ of 100 for the MSE Loss on all datasets. For the Germany dataset, baseline models are trained with a learning rate of 0.001 (as mentioned in Rojas-Campos et al. (2022)), whereas Swin-Unet models are trained with a learning rate of 0.0001. The batch size remains consistent with previous settings at 20, and all models are trained for 30 epochs. We utilize a class weight of [1, 5, 30]. For the China dataset, all models are trained with a learning rate of $10^{-4}$ for 100 epochs. The weight configuration used is [1, 15, 10].

### 5.2 RESULTS

As shown in Table 8, for the Korea dataset, our method demonstrates an improvement of 6.3% in rain prediction CSI compared to the state-of-the-art (SOTA) approach, which is ConvLSTM. We highlight that CAMT achieves a remarkable 15.6% improvement in heavy rain prediction CSI over the NWP method, which is the first DL model to surpass NWP results for extreme weather conditions. This result underscores the potential and efficacy of data-driven methods in advancing precipitation forecasting.

For the Germany dataset, U-Net emerges as the top performer among previous models, particularly excelling in rain CSI. Notably, our method achieves a 4.7% improvement over U-Net. When it comes to heavy rain prediction, U-Net's performance is limited and the NWP model outperforms all previous DL models. Our method shows a substantial 17.4% improvement over NWP, marking a significant advancement.

In the case of the China dataset, the NWP method demonstrates better performance in both rain and heavy rain prediction compared to previous DL models. Our method achieves improvements of 26.8% and 31.8% over the NWP method under these two conditions, respectively. Previous DL methods might be struggling with this dataset due to its small sample size, while our method manages to achieve substantial improvements using the proposed CAM and multi-task training framework. This underscores the robustness and versatility of our approach. We make this dataset avail-

Table 2: Experimental Results on the proposed PostRainBench. Each model undergoes three runs with different random seeds, and we report the mean, standard deviation (std), and best performance in terms of CSI and HSS. The best results are highlighted in **bold**, with the second-best results underlined. We report the relative improvement of our method (Swin-Unet+CAMT) over the best result among the baselines and NWP. In the context of the results, '↑' indicates that higher scores are better.

| | | Rain | | | | Heavy Rain | | | |
|---|---|---|---|---|---|---|---|---|---|
| | | CSI↑ | | HSS↑ | | CSI↑ | | HSS↑ | |
| | | Mean(Std) | Best | Mean(Std) | Best | Mean(Std) | Best | Mean(Std) | Best |
| Korea | NWP | 0.263(±0.000) | | * | | 0.045(±0.000) | | * | |
| | U-Net | 0.300 (±0.025) | 0.322 | **0.384**(±0.025) | **0.408** | 0.006(±0.005) | 0.010 | 0.011(±0.009) | 0.018 |
| | ConvLSTM | 0.302 (±0.009) | 0.312 | **0.384**(±0.009) | 0.395 | 0.009(±0.007) | 0.015 | 0.016(±0.012) | 0.026 |
| | MetNet | 0.298 (±0.012) | 0.307 | 0.375(±0.014) | 0.384 | 0.005(±0.007) | 0.012 | 0.009(±0.012) | 0.023 |
| | **Ours** | **0.321** (±0.005) | **0.326** | **0.384**(±0.007) | 0.389 | **0.052**(±0.010) | **0.058** | **0.089**(±0.017) | **0.097** |
| | **Ours △** | +6.3% | | +0% | | +15.6% | | +456.3% | |
| Germany | NWP | 0.338(±0.000) | | 0.252(±0.000) | | 0.178(±0.000) | | 0.173(±0.000) | |
| | U-Net | 0.491 (±0.007) | 0.495 | 0.601(±0.006) | 0.605 | 0.082(±0.028) | 0.107 | 0.148(±0.048) | 0.189 |
| | ConvLSTM | 0.477 (±0.026) | 0.478 | 0.587(±0.004) | 0.590 | 0.091(±0.041) | 0.121 | 0.162(±0.068) | 0.212 |
| | MetNet | 0.485 (±0.002) | 0.487 | 0.595(±0.005) | 0.599 | 0.027(±0.016) | 0.094 | 0.147(±0.027) | 0.168 |
| | **Ours** | **0.514** (±0.003) | **0.518** | **0.609**(±0.006) | **0.616** | **0.209**(±0.014) | **0.224** | **0.339**(±0.020) | **0.359** |
| | **Ours △** | +4.7% | | +1.3% | | +17.4% | | +96.0% | |
| China | NWP | 0.164(±0.000) | | 0.123(±0.000) | | 0.110 (±0.000) | | 0.089(±0.000) | |
| | U-Net | 0.065 (±0.007) | 0.073 | 0.093(±0.009) | 0.103 | 0.058(±0.014) | 0.070 | 0.089(±0.024) | 0.110 |
| | ConvLSTM | 0.054 (±0.011) | 0.066 | 0.079(±0.009) | 0.088 | 0.065(±0.003) | 0.068 | 0.104(±0.010) | 0.114 |
| | MetNet | 0.064 (±0.019) | 0.078 | 0.061(±0.047) | 0.106 | 0.057(±0.017) | 0.076 | 0.069(±0.057) | 0.118 |
| | **Ours** | **0.208** (±0.007) | **0.216** | **0.274**(±0.014) | **0.289** | **0.145**(±0.015) | **0.163** | **0.225**(±0.019) | **0.246** |
| | **Ours △** | +26.8% | | +122.8% | | +31.8% | | +116.3% | |

\* For Korea dataset, NWP method's HSS is not reported. For all NWP method, we only have the mean value.

able and integrate it with the previous two datasets, creating a unified benchmark that could facilitate future research in this field.

## 5.3 ABLATION STUDY

### 5.3.1 CAMT COMPONENT

We conduct an ablation study by systematically disabling certain components of our CAMT Component and evaluating the CSI results for both rain and heavy rain in Table 3 . Specifically, we focus on the weighted loss, multi-task learning, and channel attention modules as these are unique additions to the Swin-Unet backbone. In the first part, we use Swin-Unet with CAMT framework (a) as a baseline and we disable each component in CAMT and demonstrate their respective outcomes. In the second part, we use Swin-Unet without CAMT framework (e) as a baseline and we gradually add each component to the model to understand its role.

**Weigthed Loss**  (b) Without the weighted Loss in CAMT, there is a slight increase in rain CSI, but heavy rain CSI shows a dominant 97.6% decrease. (f) Adding the weighted loss to Swin-Unet results in a 6.0% decrease in rain CSI, but a significant improvement in heavy rain CSI.

**Multi-Task Learning**  (c) Without multi-task learning, there is a 3.7% drop in rain CSI, along with a notable 8.1% decrease in heavy rain CSI. (g) Incorporating multi-task learning into Swin-Unet leads to a comparable performance of rain CSI but brings a slight increase in heavy rain CSI.

**CAM**  (d) In the absence of CAM, we observe a 1.8% decrease in rain CSI and a significant 11.1% decrease in heavy rain CSI. (h) The introduction of CAM into Swin-Unet leads to a rain CSI similar to the baseline but demonstrates an impressive 11.5% improvement in heavy rain CSI. It indicates that CAM is effective for selecting and modeling multiple weather variables.

Table 3: Ablation study on Germany dataset (Rojas-Campos et al., 2022). We disable components of the framework in each experiment and report rain and heavy rain CSI as the evaluation metric.

| | Weighted Loss | Multi-Task Learning | CAM | Rain CSI↑ | Rain HSS↑ | Heavy Rain CSI↑ | Heavy Rain HSS↑ |
|---|---|---|---|---|---|---|---|
| (a) | ✓ | ✓ | ✓ | 0.514 | 0.609 | 0.209 | 0.339 |
| (b) | ✗ | ✓ | ✓ | 0.517 (+0.6%) | 0.625 (+2.6%) | 0.042 (−97.6%) | 0.008 (−11.1%) |
| (c) | ✓ | ✗ | ✓ | 0.495 (−3.7%) | 0.588 (−3.4%) | 0.192 (−8.1%) | 0.317 (−6.5%) |
| (d) | ✓ | ✓ | ✗ | 0.505 (−1.8%) | 0.602 (−1.1%) | 0.183 (−11.1%) | 0.305 (−11.1%) |
| (e) | ✗ | ✗ | ✗ | 0.521 | 0.628 | 0.000 | 0.000 |
| (f) | ✓ | ✗ | ✗ | 0.490 (−6.0%) | 0.580 (−7.6%) | 0.188 ↑↑↑ | 0.307 ↑↑↑ |
| (g) | ✗ | ✓ | ✗ | 0.516 (−0.1%) | 0.629 (+0.2%) | 0.067 ↑ | 0.007 ↑ |
| (h) | ✗ | ✗ | ✓ | 0.513 (−1.5%) | 0.624 (−0.6%) | 0.115 ↑↑ | 0.204 ↑↑ |

Although Swin-Unet can achieve a relatively high CSI when used alone (e), it does not have the ability to predict heavy rain. Importantly, these three enhancements complement each other. Weighted loss and multi-task learning are effective in improving simultaneous forecasting under the unbalanced distribution of light rain and heavy rain, while CAM provides comprehensive improvements.

### 5.3.2 ABLATION ON BACKBONE

We conduct another ablation study by replacing Swin-Unet backbone with ViT (Dosovitskiy et al., 2020) backbone under our CAMT framework in Table 4.

For the Korea dataset, ViT outperforms Swin-Unet in rain CSI and HSS but shows a slight decrease in heavy rain CSI. Importantly, its performance remains higher than that of NWP, which shows the effectiveness of CAMT. For the Germany dataset, though its performance on rain CSI is limited, the ViT model still demonstrates a remarkable performance in heavy rain CSI and surpasses NWP. For the China dataset, ViT outperforms all baseline models and is only second to Swin-Unet.

Table 4: Ablation study with ViT backbone, we highlight the best results in **bold**.

| | | Rain CSI↑ Mean(Std) | Rain CSI↑ Best | Rain HSS↑ Mean(Std) | Rain HSS↑ Best | Heavy Rain CSI↑ Mean(Std) | Heavy Rain CSI↑ Best | Heavy Rain HSS↑ Mean(Std) | Heavy Rain HSS↑ Best |
|---|---|---|---|---|---|---|---|---|---|
| Korea | ViT+CAMT | **0.326** (±0.004) | **0.329** | **0.394**(±0.001) | **0.395** | 0.049(±0.010) | 0.055 | 0.083(±0.017) | **0.097** |
| | Swin-Unet+CAMT | 0.321 (±0.005) | 0.326 | 0.384(±0.007) | 0.389 | **0.052**(±0.010) | **0.058** | **0.089**(±0.017) | **0.097** |
| Germany | ViT+CAMT | 0.484 (±0.004) | 0.488 | 0.576(±0.005) | 0.581 | 0.194(±0.023) | 0.041 | 0.050(±0.043) | 0.078 |
| | Swin-Unet+CAMT | **0.514** (±0.003) | **0.518** | **0.609**(±0.006) | **0.616** | **0.209**(±0.014) | **0.224** | **0.339**(±0.020) | **0.359** |
| China | ViT+CAMT | 0.177 (±0.004) | 0.181 | 0.217(±0.006) | 0.224 | 0.068(±0.033) | 0.105 | 0.091(±0.052) | 0.149 |
| | Swin-Unet+CAMT | **0.208** (±0.007) | **0.216** | **0.274**(±0.014) | **0.289** | **0.145**(±0.015) | **0.163** | **0.225**(±0.019) | **0.246** |

These experiments highlight the potential of the ViT model. We also conduct experiments with three baseline models but observe limited improvements. We believe that addressing the challenge of imbalanced precipitation forecasting requires a more robust backbone and the use of our CAMT framework, which incorporates multi-task information to enrich the learning process of this task.

## 6 CONCLUSION

In this paper, we introduce **PostRainBench**, a comprehensive multi-variable benchmark for NWP post-processing-based precipitation forecasting and we present **CAMT**, Channel Attention Enhanced Multi-task Learning framework with a specially designed weighted loss function. Our approach demonstrates outstanding performance improvements compared to the three baseline models and the NWP method. In conclusion, our research provides novel insights into the challenging domain of highly imbalanced precipitation forecasting tasks. We believe our benchmark could help advance the model development of the research community.

**Reproducibility Statement**  We provide the details of implementing our method as well as instructions to reproduce the experiments in Section 5. We provide the datasets we used and our code in supplementary material.

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

# A APPENDIX

## A.1 BASELINES

**U-Net** (Ronneberger et al., 2015) is a model specifically crafted to address the challenge of image segmentation in biomedical images. It excels in capturing essential features in a reduced-dimensional form during the propagation phase of its encoder component.

**ConvLSTM** (Shi et al., 2015; 2017) is a hybrid model integrating LSTM and convolutional operations. LSTMs are tailored for capturing temporal relationships, while convolutional operations specialize in modeling spatial patterns. This combination allows ConvLSTM to effectively model both temporal and spatial relationships within sequences of images.

**MetNet** (Sønderby et al., 2020) incorporates a spatial downsampler, achieved through convolutional layers, to reduce input size. Its temporal encoder employs the ConvLSTM structure, enabling the capture of spatial-temporal data on a per-pixel basis. The feature map subsequently undergoes self-attention in the Spatial Aggregator to integrate global context, before being processed by a classifier that outputs precipitation probabilities for each pixel.

**Swin-Unet** (Cao et al., 2022a) is a Unet-like Transformer. The tokenized image patches are fed into the Swin Transformer-based U-shaped Encoder-Decoder architecture with skip connections for local-global semantic feature learning. Specifically, it uses hierarchical Swin Transformer (Liu et al., 2021) with shifted windows as the encoder and decoder.

**ViT** (Dosovitskiy et al., 2020) apply a pure Transformer architecture on image data, by proposing a simple, yet efficient image tokenization strategy. We follow previous work (Tarasiou et al., 2023) to employ Transformers for dense prediction.

**FourCastNet** (Pathak et al., 2022) is a data-driven global weather forecasting model known for its rapid and accurate predictions, excelling in high-resolution forecasting of complex meteorological variables, which is based on Adaptive Fourier Neural Operators (AFNO).

## A.2 POSTRAINBENCH DATASET SUMMARY

For Korea dataset and Germany dataset variables, please refer to previous research. For China dataset variables, please refer to Table 6.

Table 5: Comparison of three NWP datasets with different spatial and temporal resolutions.

| Dataset | Korea | Germany | China |
|---|---|---|---|
| Variable type | | Pressure Level and Surface | |
| Variable numbers | 12 | 143 | 28 |
| Time period | 2020-2021 | 2011-2018 | 2020-2021 |
| Spatial resolution | 12km $\times$ 12km | 2.8km $\times$ 2.8km | 1km $\times$ 1km |
| Temporal resolution | 1h | 3h | 3h |
| Temporal Window Size | 6 | 1 | 1 |
| Data shape (T C H W) | (6, 12, 50, 65) | (1, 143, 64, 64) | (1, 28, 64, 64) |
| Data split [train val test] | [4920, 2624, 2542] | [15189, 2725, 2671] | [2264, 752, 760] |
| Data size | 47.9GB | 16.2GB | 3.6GB |

## A.3 SWIN-UNET ARCHITECTURE

The overall architecture of Swin-Unet is presented in Figure 3. In our multi-task framework, two linear projection layers are applied to output the pixel-level classification and regression predictions.

## A.4 EXPERIMENT RESULTS WITH MORE METRICS

We report more evaluation metrics of all models as follows:

Table 6: List of variables contained in the China dataset.

| Type | Long name | Short name | Level | Unit |
|---|---|---|---|---|
| Pressure Level | U-component of wind | u | 200,500,700,850,925 | $(ms^{-1})$ |
| | V-component of wind | v | 200,500,700,850,925 | $(ms^{-1})$ |
| | Temperature | T | 500,700,850,925 | (K) |
| | Relative humidity | rh liq | 500,700,850,925 | (%) |
| Surface | Rain | rain | * | $(mm/h)$ |
| | Convective Rain | rain_thud | * | $(mm/h)$ |
| | Large-scale Rain | rain_big | * | $(mm/h)$ |
| | Convective Available Potential Energy" | cape | * | $(J/kg)$ |
| | Precipitable Water | PWAT | * | $(kg/m^2)$ |
| | Mean Sea Level | msl | * | $(hPa)$ |
| | 2m temperature | t2m | * | $(^\circ C)$ |
| | 2m dew point temperature | d2m | * | $(^\circ C)$ |
| | 10m component of wind | u10m | * | $(ms^{-1})$ |
| | 10m v component of wind | v10m | * | $(ms^{-1})$ |

- **Critical Success Index (CSI)** (Donaldson et al., 1975) is a categorical metric that takes into account various elements of the confusion matrix, similar with F1-score having the value as $\frac{TP_k}{TP_k+FN_k+FP_k}$.

- **Heidke Skill Score (HSS)** (Woo & Wong, 2017) as stated by (Hogan et al., 2010), is more equitable in evaluating the forecasting performance. Higher HSS means better performance and a positive HSS indicates that a forecast is better than a random-based forecast. HSS is calculated as $\frac{2\times(TP_k\times TN_k-FN_k\times FP_k)}{FP_k^2+TN_k^2+2\times TP_k\times FN_k+(FP_k+TN_k)(TP_k+FP_k)}$.

- **Accuracy (ACC)** provides a comprehensive assessment of how accurately the model predicts outcomes across the entire dataset.

- **Probability of Detection (POD)** is a recall calculated as $\frac{TP_k}{TP_k+FP_K}$.

- **False Alarm Ratio (FAR)** (Barnes et al., 2009) represents the number of false alarms in relation to the total number of warnings or alarms, indicating the probability of false detection. It is computed as $\frac{FN_k}{TP_k+FN_k}$

- **Bias** quantifies the ratio between the observed frequency of a phenomenon and the frequency predicted by the forecasting model. $\frac{TP_k+FP_k}{TP_k+FN_k}$. If the value is greater than 1, it signifies that the forecast model predicts the occurrence more frequently than the actual phenomenon. Consequently, a bias value closer to 1 indicates a more accurate forecast.

For accuracy (Acc), our model performs lower than the baseline deep learning models but higher than NWP. However, it's important to note that accuracy may not provide realistic insights in an extremely imbalanced case. If the model predicts all instances as no-rain, it could achieve a better score. For probability of detection (Pod), our model ranks second only to NWP and outperforms all deep learning models. In terms of critical success index (CSI) and Heidke skill score (HSS), our model consistently outperforms the baseline models, as discussed earlier. The false alarm ratio (Far) measures whether the forecasting model predicts an event more frequently than it actually occurs. Our model exhibits higher but acceptable values in the rain category compared to other deep-learning models, reflecting the trade-off between enhanced forecasting ability and overforecast. In the heavy rain category, our model's bias is less than 1 and closer to 1, indicating a more accurate forecast.

A.5 COMPARISON WITH FOURCASTNET

For the Korea dataset, our model exhibits superior performance to FourCastNet in both rain CSI and heavy rain CSI metrics, with a marginal shortfall in rain HSS, where it trails by 1.8% behind FourCastNet. It is important to highlight that FourCastNet's predictive capability does not surpass that of NWP algorithms for heavy rain scenarios.

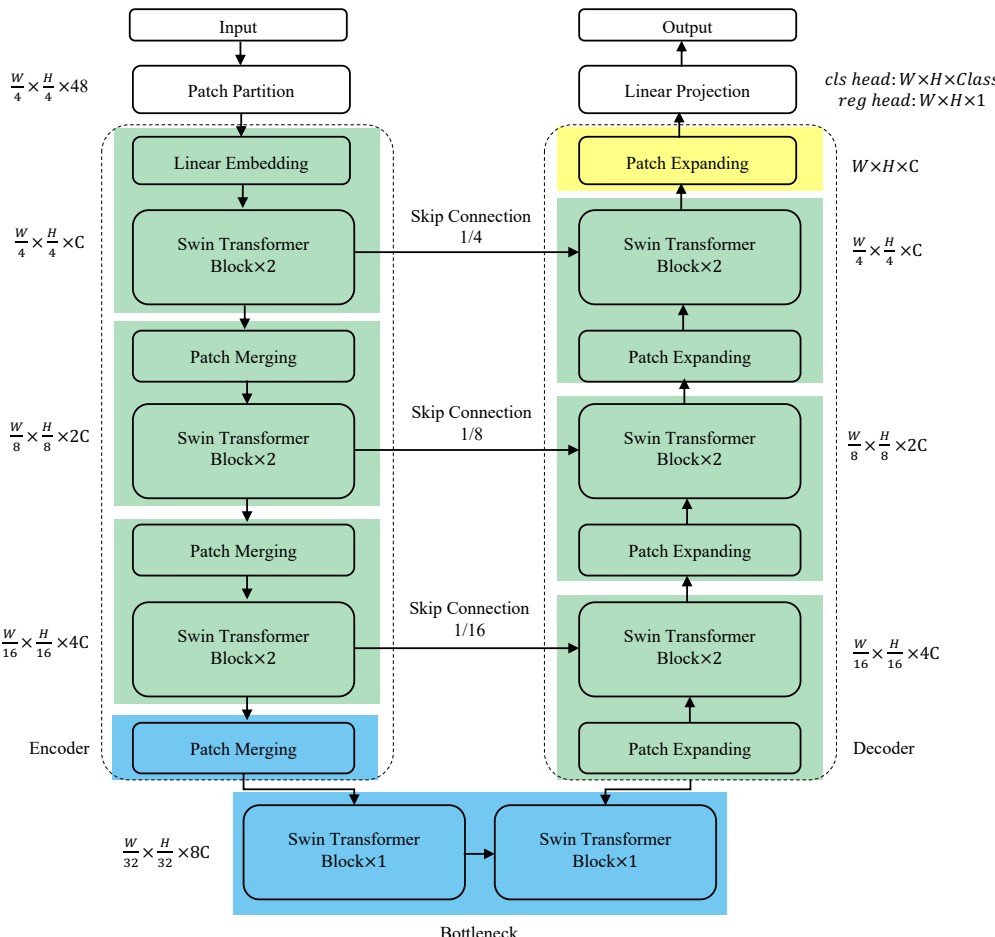

Figure 3: The architecture of Swin-Unet, which is composed of encoder, bottleneck, decoder and skip connections. Encoder, bottleneck and decoder are all constructed based on swin transformer block.

Regarding the Germany dataset, our model demonstrates an advancement over FourCastNet in all metrics for both rain and heavy rain, whereas FourCastNet does not demonstrate an advantage over NWP algorithms in heavy rain predictions.

For the China dataset, our model demonstrates comprehensive outperformance across all metrics when compared to FourCastNet. While FourCastNet posts a modest 3.6% gain over NWP methods in heavy rain forecasting, our approach achieves a substantial 31.8% improvement, marking a significant enhancement in predictive accuracy.

## A.6 PERFORMANCE ON DIFFERENT LEAD TIME ON KOREA DATASET

As shown in Figure 4, wthin the lead time interval of 6 to 20, we observe that the CSI for rain reaches a peak at a lead time of 10 before exhibiting a declining trend, whereas the CSI for heavy rain peaks at a lead time of 9, subsequently showing a fluctuating trajectory.

Expanding the analysis to a lead time range of 6 to 87, both rain and heavy rain CSI exhibit parallel trends, with heavy rain demonstrating superior performance over extended lead times, likely reflective of inherent data characteristics. Across all evaluated lead times from 6 to 87, our model's mean performance is enhanced, underscoring the comprehensive superiority of our modeling approach.

Table 7: Evaluation metrics on three datasets. Best performances are marked in **bold**. '↑' indicates that higher scores are better, '↓' indicates that higher scores are worse.

| | | Rain | | | | | | Heavy Rain | | | | | |
|---|---|---|---|---|---|---|---|---|---|---|---|---|---|
| | | Acc↑ | POD↑ | CSI↑ | FAR↓ | Bias | HSS↑ | Acc↑ | POD↑ | CSI↑ | FAR↓ | Bias | HSS↑ |
| Korea | NWP | 0.747 | **0.633** | 0.263 | 0.690 | 2.042 | * | 0.985 | 0.055 | 0.045 | **0.795** | 0.266 | * |
| | U-Net | **0.860** | 0.430 | 0.305 | **0.489** | 0.841 | 0.387 | **0.987** | 0.001 | 0.001 | 0.750 | 0.002 | 0.001 |
| | ConvLSTM | **0.860** | 0.446 | 0.312 | 0.492 | 0.878 | **0.395** | 0.986 | 0.011 | 0.010 | 0.874 | 0.083 | 0.018 |
| | MetNet | 0.853 | 0.457 | 0.307 | 0.517 | 0.946 | 0.384 | **0.987** | 0.013 | 0.012 | 0.805 | 0.067 | 0.023 |
| | Ours | 0.832 | 0.559 | **0.322** | 0.569 | 1.299 | 0.388 | 0.979 | **0.067** | **0.048** | 0.908 | 0.729 | **0.068** |
| Germany | NWP | 0.728 | **0.925** | 0.338 | 0.652 | 2.657 | 0.252 | 0.980 | **0.434** | 0.178 | 0.767 | 1.863 | 0.173 |
| | U-Net | **0.903** | 0.631 | 0.495 | **0.305** | 0.908 | 0.605 | **0.990** | 0.053 | 0.051 | **0.412** | 0.090 | 0.095 |
| | ConvLSTM | 0.896 | 0.623 | 0.475 | 0.334 | 0.935 | 0.583 | **0.990** | 0.048 | 0.045 | 0.566 | 0.111 | 0.085 |
| | MetNet | 0.895 | 0.653 | 0.483 | 0.349 | 1.003 | 0.590 | **0.990** | 0.000 | 0.000 | 0.694 | 0.001 | 0.001 |
| | Ours | 0.884 | 0.811 | **0.513** | 0.418 | 1.393 | **0.610** | 0.989 | 0.280 | **0.207** | 0.557 | 0.632 | **0.338** |
| China | NWP | 0.843 | 0.433 | 0.164 | 0.792 | 2.082 | 0.123 | 0.903 | **0.348** | 0.110 | 0.861 | 2.512 | 0.089 |
| | U-Net | **0.914** | 0.071 | 0.060 | 0.725 | 0.261 | 0.084 | **0.950** | 0.053 | 0.042 | 0.821 | 0.294 | 0.064 |
| | ConvLSTM | 0.909 | 0.083 | 0.066 | 0.756 | 0.339 | 0.088 | 0.941 | 0.099 | 0.066 | 0.837 | 0.607 | 0.094 |
| | MetNet | 0.915 | 0.086 | 0.072 | **0.680** | 0.268 | 0.106 | 0.947 | 0.104 | 0.076 | 0.778 | 0.466 | 0.118 |
| | Ours | 0.873 | **0.454** | **0.216** | 0.708 | 1.553 | **0.289** | 0.943 | 0.210 | **0.135** | **0.727** | 0.768 | **0.209** |

Table 8: Experiment result compared with FourCastNet. Each model undergoes three runs with different random seeds, and we report the mean, standard deviation (std), and best performance in terms of CSI and HSS. The best results are highlighted in **bold**. In the context of the results, '↑' indicates that higher scores are better.

| | | Rain | | | | Heavy Rain | | | |
|---|---|---|---|---|---|---|---|---|---|
| | | CSI↑ | | HSS↑ | | CSI↑ | | HSS↑ | |
| | | Mean(Std) | Best | Mean(Std) | Best | Mean(Std) | Best | Mean(Std) | Best |
| Korea | NWP | 0.263(±0.000) | | * | | 0.045(±0.000) | | * | |
| | FourCastNet | 0.314 (±0.016) | 0.325 | **0.391**(±0.023) | **0.409** | 0.011(±0.008) | 0.017 | 0.020(±0.014) | 0.029 |
| | **Ours** | **0.321** (±0.005) | **0.326** | 0.384(±0.007) | 0.389 | **0.052**(±0.010) | **0.058** | **0.089**(±0.017) | **0.097** |
| Germany | NWP | 0.338(±0.000) | | 0.252(±0.000) | | 0.178(±0.000) | | 0.173(±0.000) | |
| | FourCastNet | 0.494 (±0.009) | 0.504 | 0.595(±0.009) | 0.601 | 0.157(±0.034) | 0.185 | 0.265(±0.051) | 0.306 |
| | **Ours** | **0.514**(±0.003) | **0.518** | **0.609**(±0.006) | **0.616** | **0.209**(±0.014) | **0.224** | **0.339**(±0.020) | **0.359** |
| China | NWP | 0.164(±0.000) | | 0.123(±0.000) | | 0.110 (±0.000) | | 0.089(±0.000) | |
| | FourCastNet | 0.163 (±0.006) | 0.167 | 0.219(±0.010) | 0.230 | 0.114(±0.013) | 0.129 | 0.166(±0.023) | 0.192 |
| | **Ours** | **0.208** (±0.007) | **0.216** | **0.274**(±0.014) | **0.289** | **0.145**(±0.015) | **0.163** | **0.225**(±0.019) | **0.246** |

\* For Korea dataset, NWP method's HSS is not reported. For all NWP method, we only have the mean value.

## A.7 VALIDATION LOSS ON GERMANY DATASET

In our ablation study, we visualized the validation loss for different configurations of our model on the Germany Dataset to assess the impact of each proposed component. The validation loss curve for the standalone SwinUnet displayed an upward trend, suggesting a potential for overfitting or an insufficient capture of the dataset's essential patterns. Conversely, the integration of our proposed Channel Attention Module (CAM) and Weighted Loss (WL) resulted in a downward trend of the loss over epochs, indicating effective learning of the data distribution and improved generalizability of the model.

The CAM, with its targeted focus on salient features, and the WL, which addresses class imbalance, have shown a discernible positive influence on the model's learning process, as demonstrated by a consistent reduction in validation loss. This reduction substantiates our method's capability to tackle the specific challenges associated with precipitation forecasting in imbalanced datasets.

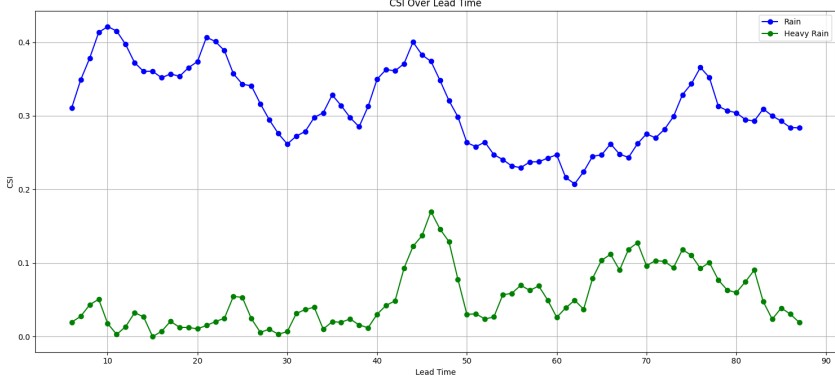

Figure 4: CSI scores of Korea Dataset for rain and heavy rain classification with lead times ranging from 6 to 87 hours.

Ultimately, the depicted loss curves validate our method's proficiency in grasping the complexities of the forecasting task, where the integrated components not only counteract overfitting but also significantly bolster the model's forecasting accuracy.

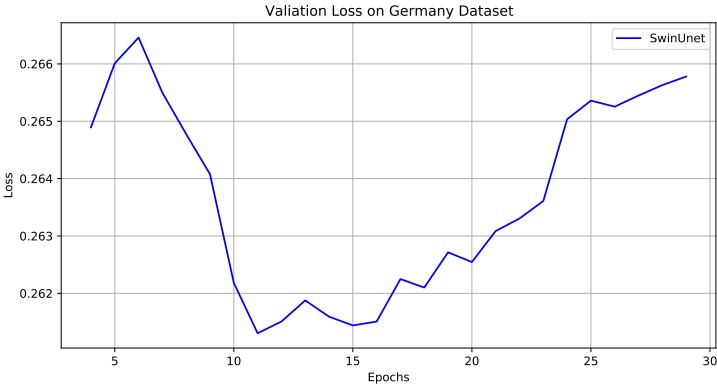

Figure 5: Valiation loss on Germany Dataset with SwinUnet.

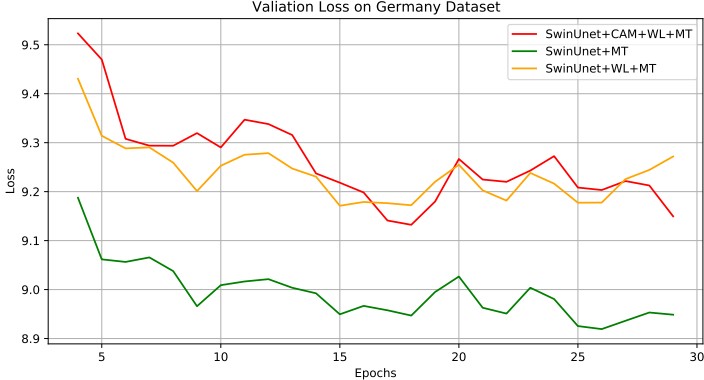

Figure 6: Validation loss on Germany Dataset with SwinUnet and proposed components: CAM (Channel Attention Module), WL (Weighted Loss), and MT (Multi-task Learning).

