# OpenReview forum: "PostRainBench: A Comprehensive Benchmark and A New Model for Precipitation Forecasting"
_ICLR.cc/2024/Conference — Submitted to ICLR 2024_

### Official Review · Reviewer_M7MM · 2023-10-27

**Soundness:** 2 fair
**Presentation:** 1 poor
**Contribution:** 2 fair
**Rating:** 5
**Confidence:** 5

**Summary:**

Precipitation forecasting holds significant scientific and societal value. While data-driven techniques are increasingly popular in addressing forecasting challenges, they struggle with accurately representing the physics involved. Marrying AI post-processing with established Numerical Weather Prediction (NWP) methods can bolster forecast accuracy. Yet, predicting heavy rainfall remains a hurdle, given uneven precipitation data and intricate meteorological variable interplays. This study presents the PostRainBench, a robust NWP post-processing benchmark comprising three datasets for enhanced precipitation forecasting. We introduce CAMT, a Channel Attention Enhanced Multi-task Learning framework, integrated with a tailored weighted loss function. This framework seamlessly merges with various backbones. Tests on our benchmark indicate that CAMT surpasses existing methods by notable margins, especially under extreme rainfall conditions. Significantly, our model is the pioneering deep learning tool that outdoes traditional NWP methods in predicting intense rain, showcasing its immense promise in mitigating extreme weather implications.

**Strengths:**

The introduction of the CAMT framework stands out as a noteworthy achievement. Its ability to integrate smoothly with different backbones offers versatility, while its significant performance improvements in extreme precipitation predictions highlight its potential in addressing some of the most challenging aspects of weather forecasting.

**Weaknesses:**

1. The authors claim in the paper that they are the first to surpass NWP using a deep learning approach. However, to my knowledge, several studies have already ventured into this area, as evidenced by references [1-3]. I suggest the authors review the accuracy of this statement and make appropriate amendments.

2. In my perspective, the proposed model in this paper seems to lack innovation, appearing to be a combination of the swinTransformer and unet. This raises concerns about the reproducibility of the experimental results. Could the authors provide more details or rationale to support their design choices?

3. The selected baselines in the paper don't appear comprehensive. For instance, recent significant works in the relevant field, such as OpenSTL[4] and Fourcastnet[5], are not considered. I would recommend the authors to include these for comparison.


[1] Zhang, Yuchen, et al. "Skilful nowcasting of extreme precipitation with NowcastNet." *Nature* 619, no. 7970 (2023): 526-532.

[2] Bi, Kaifeng, et al. "Pangu-weather: A 3d high-resolution model for fast and accurate global weather forecast." *arXiv preprint arXiv:2211.02556* (2022).

[3] Chen, Kang, et al. "FengWu: Pushing the Skillful Global Medium-range Weather Forecast beyond 10 Days Lead." *arXiv preprint arXiv:2304.02948* (2023).

[4] Tan, Cheng, et al. "OpenSTL: A Comprehensive Benchmark of Spatio-Temporal Predictive Learning." *arXiv preprint arXiv:2306.11249* (2023).

[5] Pathak, Jaideep, et al. "Fourcastnet: A global data-driven high-resolution weather model using adaptive fourier neural operators." *arXiv preprint arXiv:2202.11214* (2022).

**Questions:**

See Weaknesses

---

> ### Author Response · Authors · 2023-11-22
> **Response to Reviewer M7MM-Part1**
>
> Q4.1: Claim of the first to surpass NWP.
>
> A4.1:
>
> We respectfully point out the key misunderstanding. To clarify, our assertion is that our model is the first to surpass NWP in the specific context of precipitation forecasting, particularly in extreme cases, within the realm of NWP post-processing tasks. This claim is based on three key criteria:
>
> 1. Focus on Precipitation Forecasting: Our model specifically addresses precipitation forecasting, which distinguishes it from other studies. While recent models like Pangu [2] and FengWu [3] forecast various upper-air and surface weather variables, they do not include precipitation in their scope. Our work fills this gap by providing insights specifically for precipitation forecasting, contributing significant advancements in this area.
> 2. NWP Post-processing Task: Our approach uniquely combines physics-based NWP results with AI methods in a post-processing task, a methodology that diverges from the purely data-driven deep learning methods employed in studies like Pangu and FengWu. This hybrid approach is in line with the future direction suggested in the Pangu study, which anticipates the combination of AI-based and NWP methods for improved performance.
>
> 3. Excellence in Extreme Cases: The specific area where our model excels is in forecasting heavy rain, which we define as 'extreme cases' in our paper. This is a crucial distinction, as none of the deep learning models in the related work, including KoMet, have surpassed NWP in this particular aspect of heavy rain forecasting.
>
> Regarding NowcastNet [1], while it represents a significant advancement in using radar observations for weather prediction, its methodological framework differs from ours. NowcastNet relies on a data-driven approach without integrating physics-informed predictions (NWP Method), whereas our model is designed specifically for NWP post-processing, leveraging both AI and physics-based insights.
>
> In summary, our claim of being the first to surpass NWP in the specific context of heavy rain forecasting in NWP post-processing tasks is based on these distinct aspects of our work. We believe this clarification underscores the unique contribution of our study in advancing the field of precipitation forecasting.
>
> Q4.2: Novelty of the method and rationale of design choices
>
> A4.2:
>
> Please also refer to A2.1 for Reviewer Lop7.
>
> We understand the reviewer's concerns regarding the perceived novelty of our model. It's true that our model integrates well-known elements like the Swin Transformer and Unet, creating what is known as SwinUnet. However, the innovation in our work extends beyond this integration, focusing on the unique application of these components to meet specific challenges in NWP post-processing tasks.
>
> Channel Attention Module (CAM) and Multi-task Learning: A central innovative aspect of our approach is the integration of CAM with multi-task learning. This combination specifically addresses the issue of imbalanced data distribution in weather forecasting. By selectively focusing on crucial weather variables through CAM, our model achieves enhanced predictive accuracy. Concurrently, multi-task learning offers a nuanced approach to managing the complexities inherent in various aspects of precipitation forecasting.
>
> Innovative Application of SwinUnet for Precipitation Forecasting: Our major contribution lies in the novel application of the SwinUnet architecture, adapted to address real-world challenges in precipitation forecasting. The Swin Transformer, known for its effectiveness in feature extraction within the realm of computer vision, is here repurposed in a hierarchical transformer-based method tailored for NWP post-processing tasks. This marks a significant first in applying such an architecture to this specific area.
>
> The task itself necessitated an innovative approach, leading us to adopt state-of-the-art models like SwinUnet to effectively tackle the challenges of NWP post-processing. Our model isn't just a mere combination of existing technologies; it's a tailored solution addressing an under-explored and crucial area in weather forecasting.
>
> In summary, the innovation of our model is rooted in its application and the substantial improvements it brings to precipitation forecasting. This includes addressing the challenge of imbalanced data and enhancing the robustness of forecasting models, thereby contributing valuable advancements to the field.
>
> In order to ensure the reproducibility of our research, we have included both the code and the relevant data in the supplementary materials.

---

> > ### Comment · Reviewer_M7MM · 2023-11-23
> >
> > Thank you for your response, however, I don't think previous models can't predict precipitation, either pangu-weather or nowcastnet; there is still skepticism as to whether or not they were the first to do so. nowcastnet itself is supposed to be a data-driven model that incorporates physical information, and the original article mentions incorporating the advection-diffusion equations, so why do you say that he didn't incorporate physical information? I would suggest rewriting and changing the paper to restate the adjustments to the solution.

---

> > > ### Author Response · Authors · 2023-11-23
> > > **Response to Reviewer M7MM and Looking forward to your reply**
> > >
> > > Thank you for your insightful comments. We would like to answer your questions below.
> > >
> > > Q4.4 Can previous models Pangu-weather or Nowcastnet predict precipitation?
> > >
> > > A4.4
> > >
> > > We would like to clarify the distinctions among different weather forecasting methodologies to address the concerns raised.
> > >
> > > 1. **Medium-range Weather Forecasting:** Models like Pangu-weather and Fengwu engage in medium-range forecasting, utilizing **ERA5 Reanalysis data** for both input and output. Our investigation has determined that these models do not research on precipitation.
> > > 2. **Precipitation Nowcasting:** NowcastNet, for instance, is geared towards nowcasting, using **radar data**. It does integrate physical information via advection-diffusion equations, but its input and output variable is **radar data** with precipitation single variable.
> > > 3. **NWP Post-processing:** This is where our work resides, focusing on bias correction of forecasts(defined in [1,2]) and combining AI with **NWP predictions**(defined in [3]). Unlike NowcastNet, our model uses multiple surface and upper-air variables, including precipitation, to refine the NWP output. This multi-variable approach is essential for understanding the interplay between different weather elements, which is central to our task.
> > >
> > > Our claim of innovation pertains to surpassing NWP methods in NWP Post-processing, not precipitation nowcasting. This is based on the most closely related work, Komet[4], and the unique challenges addressed by our model.
> > >
> > > Q4.5 Whether NowcastNet incorporates physical information?
> > >
> > > A4.5
> > >
> > > I would like to state that our claim is "NowcastNet relies on a data-driven approach without integrating **physics-informed predictions** (NWP Method)". We emphasize **predictions** on the data side.
> > >
> > > Yes, NowcastNet incorporates physical information by using advection-diffusion equations on **the model side**. However, our task differentiates itself by not only incorporating physical laws through NWP predictions from **the data side** but also employing a deep learning model that accounts for the relationships between multiple weather variables, which is critical for NWP Post-processing.
> > >
> > > We are grateful for the opportunity to discuss these points and hope this clarifies the positioning and contributions of our work.
> > >
> > > [1] Han, Lei, et al. "A deep learning method for bias correction of ECMWF 24–240 h forecasts." *Advances in Atmospheric Sciences* 38.9 (2021): 1444-1459.
> > >
> > > [2] Reichstein, Markus, et al. "Deep learning and process understanding for data-driven Earth system science." *Nature* 566.7743 (2019): 195-204.
> > >
> > > [3] Bi, K., Xie, L., Zhang, H., Chen, X., Gu, X., & Tian, Q. (2023). Accurate medium-range global weather forecasting with 3D neural networks. Nature, 619(7970), 533-538.
> > >
> > > [4] Kim, Taehyeon, et al. "Benchmark Dataset for Precipitation Forecasting by Post-Processing the Numerical Weather Prediction." *arXiv preprint arXiv:2206.15241* (2022).

---

> ### Author Response · Authors · 2023-11-22
> **Response to Reviewer M7MM-Part2**
>
> Q4.3: Comparison with recent significant works
>
> A4.3:
>
> We appreciate the reviewer's suggestion regarding the inclusion of comprehensive baselines in our paper. While OpenSTL [4] represents a comprehensive benchmark for spatiotemporal forecasting, our work focuses on NWP post-processing tasks, which differ from the typical spatiotemporal prediction task of inputting N frames to output N frames.
>
> In our benchmark, we have defined two specific tasks:
>
> 1. Sequence to Image Prediction (Korea Dataset): Here, we input N frames to predict 1 frame under fixed lead times varying from 6 to 87 hours.
>
> 2. Image-to-Image Translation Prediction (Germany and China Datasets): This involves inputting 1 frame to predict the future frame under a fixed 3-hour lead time. Notably, these experiments, especially on the latter two datasets, do not involve temporal information.
> As such, our NWP post-processing tasks are distinct from the typical input-output frame prediction models used in OpenSTL [4]. This difference in task nature and requirements is why we have not included a comparison with the spatiotemporal forecasting methods in OpenSTL in our study.
>
> In selecting our baselines, we choose models like ConvLSTM, which is recurrent-based, and U-Net, which is recurrent-free, along with the MetNet network, specifically designed for precipitation forecasting. Our proposed SwinUnet model is also a recurrent-free method.
> Acknowledging the reviewer's valuable input, we have included additional experiments with the latest FourCastNet as a baseline. We maintained consistent settings with baseline models for the respective dataset and averaged results over three random seeds.
>
> We have supplemented this result in Section A.5 in the appendix.
>
> Korea Dataset
>
> |             | CSI_Rain(Mean/Std) | CSI_Rain(Best) | HSS_Rain(Mean/Std) | HSS_Rain(Best) | CSI_Heavy(Mean/Std) | CSI_Heavy(Best) | HSS_Heavy(Mean/Std) | HSS_Heavy(Best) |
> | :---------- | :----------------- | :------------- | :----------------- | :------------- | :------------------ | :-------------- | :------------------ | :-------------- |
> | FourCastNet | 0.314(0.016)       | 0.325          | **0.391(0.023)**   | **0.409**      | 0.011(0.008)        | 0.017           | 0.020(0.014)        | 0.029           |
> | Ours        | **0.321(0.005)**   | **0.326**      | 0.384(0.007)       | 0.389          | **0.052(0.010)**    | **0.058**       | **0.089(0.017)**    | **0.097**       |
>
> Germany Dataset
>
> |             | CSI_Rain(Mean/Std) | CSI_Rain(Best) | HSS_Rain(Mean/Std) | HSS_Rain(Best) | CSI_Heavy(Mean/Std) | CSI_Heavy(Best) | HSS_Heavy(Mean/Std) | HSS_Heavy(Best) |
> | :---------- | :----------------- | :------------- | :----------------- | :------------- | :------------------ | :-------------- | :------------------ | :-------------- |
> | FourCastNet | 0.494(0.009)       | 0.504          | 0.595(0.009)       | 0.601          | 0.157(0.034)        | 0.185           | 0.265(0.051)        | 0.306           |
> | Ours        | **0.514(0.003)**   | **0.518**      | **0.609(0.006)**   | **0.616**      | **0.209(0.014)**    | **0.224**       | **0.339(0.020)**    | **0.359**       |
>
> China Dataset
>
> |             | CSI_Rain(Mean/Std) | CSI_Rain(Best) | HSS_Rain(Mean/Std) | HSS_Rain(Best) | CSI_Heavy(Mean/Std) | CSI_Heavy(Best) | HSS_Heavy(Mean/Std) | HSS_Heavy(Best) |
> | :---------- | :----------------- | :------------- | :----------------- | :------------- | :------------------ | :-------------- | :------------------ | :-------------- |
> | FourCastNet | 0.163(0.006)       | 0.167          | 0.219(0.010)       | 0.230          | 0.114(0.013)        | 0.129           | 0.166(0.023)        | 0.192           |
> | Ours        | **0.208(0.007)**   | **0.216**      | **0.274(0.014)**   | **0.289**      | **0.145(0.015)**    | **0.163**       | **0.225(0.019)**    | **0.246**       |
>
> For the Korea dataset, our model exhibits superior performance to FourCastNet in both rain CSI and heavy rain CSI metrics, with a marginal shortfall in rain HSS, where it trails by 1.8% behind FourCastNet. It is important to highlight that FourCastNet's predictive capability does not surpass that of NWP algorithms for heavy rain scenarios.
>
> Regarding the Germany dataset, our model demonstrates an advancement over FourCastNet in all metrics for both rain and heavy rain, whereas FourCastNet does not demonstrate an advantage over NWP algorithms in heavy rain predictions.
>
> For the China dataset, our model demonstrates comprehensive outperformance across all metrics when compared to FourCastNet. While FourCastNet posts a modest 3.6% gain over NWP methods in heavy rain forecasting, our approach achieves a substantial 31.8% improvement, marking a significant enhancement in predictive accuracy

---

> ### Author Response · Authors · 2023-11-23
> **Looking forward to your reply**
>
> Dear Reviewer M7MM,
>
> Thank you very much again for the time and effort put into reviewing our paper. We believe that we have addressed all your concerns in our response. We have also followed your suggestion to improve our paper and have added additional experimental analysis. We kindly remind you that we are approaching the end of the discussion period. We would love to know if there is any further concern, additional experiments, suggestions, or feedback, as we hope to have a chance to reply before the discussion phase ends.
>
> Best regards,
>
> All authors

---

### Official Review · Reviewer_aiFF · 2023-10-30

**Soundness:** 3 good
**Presentation:** 3 good
**Contribution:** 2 fair
**Rating:** 6
**Confidence:** 4

**Summary:**

Tree datasets are grouped to provide an evaluation dataset for precipitation forecasting. A method is proposed for precipitation forecasting as a weighted combination of three different algorithms with an without learning. Comparative experiments are proposed and discussed as well as an ablation study. The datasets is announce to be released after publication.

**Strengths:**

The dataset is useful and may help researchers in their work on the subject of precipitation forecasting. The obtained results in precipitation forecasting with the combination of three algorithms is significant. The comparison with other methods seems well performed.

**Weaknesses:**

The proposed algorithm is not particularly surprising.

**Questions:**

Is there a part of the datasets newly provided by the authors ?

---

> ### Author Response · Authors · 2023-11-22
> **Response to Reviewer aiFF**
>
> Q3.1: The proposed algorithm is not particularly surprising.
>
> A3.1:
>
> Please also refer to A2.1 for Reviewer Lop7.
>
> We would like to emphasize our significant contribution to an under-explored task in NWP post-processing and highlight the novel contributions of our work, particularly in the context of multi-task learning and the Channel Attention Module (CAM).
>
> Our algorithm stands out by providing a straightforward yet efficient solution to this specific challenge. It navigates the intricacies of imbalanced data scenarios, offering novel insights into optimizing models under such conditions. This approach is particularly relevant given the nature of precipitation data, which inherently exhibits significant class imbalances.
>
> In summary, the value of our algorithm lies in its application-oriented focus, offering a practical and effective solution to a well-known challenge in precipitation forecasting. This contribution provides valuable insights for the field and can serve as a foundation for future advancements in this area.
>
> Q3.2: Is there a part of the datasets newly provided by the authors?
>
> A3.2:
>
> In our study, we introduce the China Dataset, which is a novel addition provided by us. This dataset offers unique insights specific to the Chinese region's precipitation patterns, contributing to the diversity and comprehensiveness of precipitation forecasting research.
>
> While the Korea and Germany datasets used in our study have been utilized in previous research, we bring a new perspective to these existing datasets. Our comprehensive benchmarking approach includes a detailed comparison of different settings in temporal information and variable selection. This analysis not only enhances the understanding of these datasets but also extends their applicability.
>
> The significance of our contribution lies in the novel application and analysis of these datasets, particularly in how we compare and contrast different modeling approaches. This benchmarking approach is valuable for wider downstream tasks, offering insights that can be adapted and applied in various contexts beyond the specific datasets we used.
>
> In summary, our study not only introduces a new dataset with the China Dataset but also enriches the research landscape by providing a deeper and more nuanced analysis of existing datasets, broadening their potential applications in the field of precipitation forecasting.
>
> We have provided the download link of all datasets we used in the supplementary material.

---

> > ### Comment · Reviewer_aiFF · 2023-11-22
> > **Thanks**
> >
> > Thanks for your answers.

---

### Official Review · Reviewer_Lop7 · 2023-10-31

**Soundness:** 3 good
**Presentation:** 3 good
**Contribution:** 2 fair
**Rating:** 3
**Confidence:** 4

**Summary:**

The paper introduces a unique deep learning-based post-processing technique designed for Numerical Weather Prediction (NWP) methods, focusing specifically on the task of precipitation forecasting. The authors curate three distinct datasets from Korea, Germany, and China for evaluation purposes. The architecture of the proposed model is tripartite: it consists of a Channel Attention Model (CAM) to handle the high dimensionality in NWP variables, a Swin-Unet backbone, and a multi-task learning loss function that incorporates both classification and regression. Noteworthy are the contributions of the CAM and the multi-task learning loss. Comparative evaluations demonstrate that the proposed model surpasses traditional methods in performance.

**Strengths:**

1. The model delivers a marked performance improvement across all three test datasets.

2. The paper investigates an intriguing application of machine learning, namely, precipitation prediction.

**Weaknesses:**

1. The model lacks substantial innovation; both the Channel Attention Model (CAM) and the multi-task learning loss function appear to be straightforward engineering optimizations rather than novel contributions.

2. Despite acknowledging the presence of significant data imbalance, the authors do not incorporate any mechanisms within the model to address this issue.

3. The experimental section could benefit from an in-depth discussion analyzing how the model's performance varies with different lead times.

**Questions:**

Could the authors elaborate on the weaknesses in the model, particularly in relation to handling imbalanced data distributions?

---

> ### Author Response · Authors · 2023-11-22
> **Response to Reviewer Lop7-Part1**
>
> Q2.1: Novelty of the proposed method.
>
> A2.1:
> Firstly, We would like to highlight the novel contributions of our work in task formulation.
>
>  In addressing the reviewer's comment on the innovation aspect of our model, we would like to emphasize our significant contribution to an under-explored task in NWP post-processing. Our comprehensive benchmark introduces a novel approach to this task, encompassing two distinct settings: image-to-image translation with fixed lead time, and sequence-to-image under the same fixed lead time conditions.
>
> The innovative essence of our work lies in its problem-solving orientation. Based on our prior observations, despite deep learning models surpassing NWP in general rainfall prediction, a gap remained in their ability to outperform NWP models in forecasting heavy precipitation. Our proposed solution specifically addresses this challenge. We have supplemented our research with experimental results from the state-of-the-art network, FourCastNet, which also could not achieve better forecasting performance than NWP in heavy rain. This inability of existing advanced models to surpass NWP in heavy rain forecasting highlights the novelty and significance of our contribution.
>
> Our work provides key insights into resolving this critical issue in precipitation forecasting, offering a pragmatic and effective approach to enhance predictive performance in extreme weather conditions. This not only demonstrates our model's innovative application but also its potential impact and value in real-world meteorological scenarios.
>
> Secondly, we would like to highlight the novel contributions of our work, particularly in the context of multi-task learning and the Channel Attention Module (CAM):
>
> 1. Innovative Application of Multi-task Learning in Precipitation Forecasting: While multi-task learning is indeed a known optimization method, our research takes a unique approach by being the first to utilize regression to aid classification in data-imbalanced precipitation forecasting. Drawing inspiration from recent research [1] which suggests adding a classification loss in imbalanced data scenarios for regression tasks, our study extends this concept in a novel direction. We propose that for classification tasks with imbalanced data, integrating a regression loss can be beneficial. This approach deviates from traditional radar-based nowcasting tasks that predominantly rely on weighted loss for regression. Our contribution lies in introducing a new perspective to the optimization problem, potentially benefiting a broader spectrum of research within this domain.
>
> 2. Pioneering Use of CAM in Weather Variable Relation Modeling: Regarding the Channel Attention Model (CAM), our work stands out as the first to apply this mechanism to the complex task of modeling relationships between weather variables. Unlike general computer vision tasks, which often assume channel independence, our research bridges the gap between computer vision and weather forecasting by providing an innovative application of CAM. This integration is a significant step forward in enhancing the interpretability and efficacy of weather prediction models.
>
> Reference:
>
> [1] Pintea, S. L., Lin, Y., Dijkstra, J., & van Gemert, J. C. (2023). A step towards understanding why classification helps regression. In Proceedings of the IEEE/CVF International Conference on Computer Vision (pp. 19972-19981).
>
> Q2.2: Mechanisms within model to address the data imbalance issue.
>
> A2.2:
>
> We understand the reviewer's concern regarding the data imbalance issue and the necessity to address it within our model. As mentioned in Sections 4.1 and 4.2 of our paper, our approach to mitigating the impact of data imbalance involves two key strategies: weighted loss and multi-task learning.
>
> ●Weighted Loss: We employ a weighted loss mechanism to counterbalance the disproportionate representation of classes in our dataset. This method assigns greater importance to underrepresented classes, ensuring they have a significant impact on the model's learning process. However, we recognize that a simple weighted loss approach can sometimes lead to over-forecasting in smaller classes.
>
> ●Multi-task Learning: To alleviate the over-forecasting issue, we integrate multi-task learning into our framework. This technique not only addresses the imbalanced nature of the dataset but also improves the model's generalization capabilities by learning shared representations across multiple tasks. Multi-task learning provides a more holistic approach to managing data imbalance, enhancing the model's performance and predictive accuracy.

---

> ### Author Response · Authors · 2023-11-22
> **Response to Reviewer Lop7-Part2**
>
> Q2.3: Experiments of different lead times.
>
> A2.3:
>
> We have supplemented this in Section A.6 with additional results for lead times from 6 to 87 hours on the test set to provide a more detailed view of our model's performance over varying lead times.
>
> For the Korea dataset, we follow the setup of previous studies, selecting samples with various lead times ranging from 6 to 87 hours for our training, validation, and testing sets. The experimental results reported in our paper represent the average performance across different lead times within this 6-87 hour range.
>
> As shown in Figure 4 in the appendix of the revised PDF, within the lead time interval of 6 to 20, we observe that the CSI for rain reaches a peak at a lead time of 10 before exhibiting a declining trend, whereas the CSI for heavy rain peaks at a lead time of 9, subsequently showing a fluctuating trajectory. Expanding the analysis to a lead time range of 6 to 87, both rain and heavy rain CSI exhibit parallel trends, with heavy rain demonstrating superior performance over extended lead times, likely reflective of inherent data characteristics. Across all evaluated lead times from 6 to 87, our model's mean performance is enhanced, underscoring the comprehensive superiority of our modeling approach.
>
> For the Germany and China datasets, the lead time is set at a fixed 3 hours, corresponding to the paired format in which NWP predictions and precipitation observations are provided.
>
> In summary, while our approach adapts to the fixed 3-hour lead time in the Germany and China datasets, our comprehensive analysis and additional experiments on the Korea dataset underscore the versatility and robustness of our method across a range of lead times, affirming its applicability in varied precipitation forecasting contexts.
>
> Q2.4: Weakness particularly in handling imbalanced data distributions.
>
> A2.4:
>
> Discussion of Weakness:
>
> One limitation of our model is the empirical selection of weights for the hyperparameters. Future research could explore an adaptive loss mechanism within a multi-task learning framework.
>
> ●Empirical Selection of Weights: A key limitation lies in our approach to the empirical selection of weights for hyperparameters. This process is closely tied to the dataset's distribution, which can vary significantly in real-world scenarios. Due to this reliance, the model's performance might be overly dependent on the specific characteristics of the dataset used for training, potentially reducing its effectiveness when applied to datasets with different distributions.
>
> ●Potential for Adaptive Loss Mechanisms: Recognizing this limitation, we suggest that future research could explore the development of adaptive loss mechanisms within a multi-task learning framework. An adaptive approach would allow for more dynamic adjustment of weights based on the data distribution, potentially improving the model's handling of imbalanced datasets. This could lead to more nuanced and effective learning, especially in scenarios where data imbalances are pronounced and complex.
>
> In summary, while our current model presents a solid foundation, its approach to weight selection and handling of imbalanced data could benefit from further refinement. The exploration of adaptive loss mechanisms, particularly in conjunction with multi-task learning strategies, represents a promising avenue for enhancing the model's robustness and versatility in diverse data scenarios.

---

> ### Author Response · Authors · 2023-11-23
> **Looking forward to your reply**
>
> Dear Reviewer Lop7,
>
> Thank you very much again for the time and effort put into reviewing our paper. We believe that we have addressed all your concerns in our response. We have also followed your suggestion to improve our paper and have added additional experimental analysis. We kindly remind you that we are approaching the end of the discussion period. We would love to know if there is any further concern, additional experiments, suggestions, or feedback, as we hope to have a chance to reply before the discussion phase ends.
>
> Best regards,
>
> All authors

---

> > ### Comment · Reviewer_Lop7 · 2023-11-23
> >
> > Thank you for your reply. Nonetheless, I remain unconvinced about the substantial impact of the task itself on heavy rain forecasting from NWP. A more technical innovation is anticipated for a prestigious ML conference. Furthermore, a 3-hour lead time is not an interesting time zone for NWP post-processing, as NWP data is typically employed for lead times exceeding 6 hours. For example, MetNet2 leverages NWP for enhancing performance over longer lead times.

---

> > > ### Author Response · Authors · 2023-11-23
> > > **Response to Reviewer Lop7 and Looking forward to your reply**
> > >
> > > Thank you for your insightful comments.
> > >
> > > Q2.5 Concerns on the substantial impact of the task on heavy rain forecasting from NWP.
> > >
> > > A2.5
> > >
> > > We appreciate the opportunity to clarify the practical impact of our task. The motivation for our paper is derived from a bias correction challenge organized this year. This challenge was formulated by the **meteorological bureaus**, with input from **geoscience experts** and **weather forecasting companies**, underscoring its relevance and business value.
> > >
> > > We firmly believe in the task's operational worth, particularly its application to real-world scenarios for correcting heavy rain forecasts. Our approach has demonstrated its potential to effect tangible improvements in forecasting practices.
> > >
> > > The engagement of various weather forecasting entities in defining this challenge testifies to the significance and applicability of the task at hand. It is our conviction that our research contributes to addressing real-world challenges in meteorological forecasting, particularly in the critical area of heavy rain prediction.
> > >
> > > Q2.6 Concerns about Technical Innovation
> > >
> > > A2.6
> > >
> > > We appreciate the reviewer's focus on technical innovation. Our Channel Attention and Multi-Task (CAMT) Framework has been intentionally designed to balance simplicity with efficacy. Its architecture, while deceptively straightforward, is carefully crafted to adeptly tackle the complex task of enhancing the accuracy of heavy rain forecasts. By strategically integrating channel attention with multi-task learning, the CAMT Framework effectively captures and emphasizes critical features within imbalanced datasets, a substantial challenge in meteorological model prediction.
> > >
> > > This blend of simplicity and performance is particularly innovative in its application to NWP post-processing, where overly complex models can become impractical. Our approach not only advances the state-of-the-art but does so with an elegant model that facilitates easier implementation and adaptation in real-world forecasting systems.
> > >
> > > Q2.7 Concerns Regarding the 3h Lead Time
> > >
> > > A2.7
> > >
> > > 1. In the KoMet Dataset, we encompass a wide range of lead times, from **6 to 87 hours**. As detailed in Section A.6 of the appendix in our revised manuscript, our model demonstrates enhanced performance, particularly notable after a **40-hour** lead time. This evidence supports our model's capability to perform **robustly over a substantial lead time range**.
> > > 2. For the Germany and China Datasets, a 3-hour lead time aligns with operational standards in current rainfall forecasting services. Specifically, for the China Dataset, the task parameters are stipulated by local **meteorological authorities**. Moreover, the precedent of employing a 3-hour lead time in previous studies with the Germany Dataset underscores its practicality and relevance to real-world forecasting applications.
> > >
> > > These details affirm that while our model is capable of performing across various lead times, it also excels in the critical 3-hour window that is of practical importance in contemporary meteorological operations.
> > >
> > > We are grateful for the opportunity to discuss these points and hope this clarifies the positioning and contributions of our work.

---

### Official Review · Reviewer_hWvd · 2023-11-04

**Soundness:** 3 good
**Presentation:** 3 good
**Contribution:** 3 good
**Rating:** 5
**Confidence:** 5

**Summary:**

This paper constructs a comprehensive benchmark in the field of weather forecasting and proposes a baseline model to evaluate the proposed benchmark. To be specific, to facilitate the development of precipitation prediction, the authors propose to compile 3 existing precipitation datasets into a new benchmark. Meanwhile, they propose a baseline model with the designs of channel attention module, multi-task learning, weighted loss and etc.. They validate the proposed method on three datasets, and observe significant improvements over previous methods in different scenarios.

**Strengths:**

	This paper is well written and easy following.
	This paper provides all-round and detailed performance comparisons with different kinds of classical precipitation methods on these three datasets. Those comparisons will become valuable for the community if the authors release their codebases and benchmarks in the future.
	The proposed method achieves significant performance improvements over those classical precipitation methods, showing the superiority of their method.

**Weaknesses:**

1. It is suggested that the authors could show some qualitative results to demonstrate the superiority of their method. It is interesting to see how the proposed channel attention and weighted loss work, which plays a key role in this paper.
2. It is suggested that the authors could discuss some limitations of their method in this paper, and open some potential possibilities with their benchmark in this paper. I do not see many application scenarios shown in this paper, which makes this paper sound relatively restricted.
3. The lead time in this paper is fixed as 3h. In practice, the length of lead time has significant influence on precipitation results. It would be better if the authors could exploit the lead time in their method, to show the robustness of their method.
4. For a benchmark paper, the reproduce ability is very important, especially for the climate science domain. Thus, I urge the authors to present full data and code for the reproduce ability checking.

**Questions:**

please check the weakness

---

> ### Author Response · Authors · 2023-11-22
> **Response to Reviewer hWvd-Part1**
>
> Q1.1: Qualitative results
>
> A1.1:
>
> We add Section A.7 in the appendix to visualize the validation loss for different configurations of our model on the Germany Dataset to assess the impact of each proposed component.
>
> As shown in Figure 5, the validation loss curve for the standalone SwinUnet displayed an upward trend, suggesting a potential for overfitting or an insufficient capture of the dataset's essential patterns. Conversely, the integration of our proposed Channel Attention Module (CAM) and Weighted Loss (WL) resulted in a downward trend of the loss over epochs (Figure 6), indicating effective learning of the data distribution and improved generalizability of the model.
>
> The CAM, with its targeted focus on salient features, and the WL, which addresses class imbalance, have shown a discernible positive influence on the model's learning process, as demonstrated by a consistent reduction in validation loss. This reduction substantiates our method's capability to tackle the specific challenges associated with precipitation forecasting in imbalanced datasets.
>
> Ultimately, the depicted loss curves validate our method's proficiency in grasping the complexities of the forecasting task, where the integrated components not only counteract overfitting but also significantly bolster the model's forecasting accuracy.
>
> Q1.2: Limitation and applications.
>
> A1.2:
>
> Discussion of limitations:
>
> One limitation of our model is the empirical selection of weights for the hyperparameters. Future research could explore an adaptive loss mechanism within a multi-task learning framework.
>
> ●Empirical Selection of Weights: A key limitation lies in our approach to the empirical selection of weights for hyperparameters. This process is closely tied to the dataset's distribution, which can vary significantly in real-world scenarios. Due to this reliance, the model's performance might be overly dependent on the specific characteristics of the dataset used for training, potentially reducing its effectiveness when applied to datasets with different distributions.
>
> ●Potential for Adaptive Loss Mechanisms: Recognizing this limitation, we suggest that future research could explore the development of adaptive loss mechanisms within a multi-task learning framework. An adaptive approach would allow for more dynamic adjustment of weights based on the data distribution, potentially improving the model's handling of imbalanced datasets. This could lead to more nuanced and effective learning, especially in scenarios where data imbalances are pronounced and complex.
>
> In summary, while our current model presents a solid foundation, its approach to weight selection and handling of imbalanced data could benefit from further refinement. The exploration of adaptive loss mechanisms, particularly in conjunction with multi-task learning strategies, represents a promising avenue for enhancing the model's robustness and versatility in diverse data scenarios.
>
> As for the potential applications of our benchmark and the corresponding task:
>
> 1. Our released models can be directly used to improve extreme precipitation forecasting. We have shown significant improvements of 15.6%, 17.4%, and 31.8% on the Korea, Germany, and China datasets, respectively, over NWP methods.  In the current weather forecasting landscape, NWP methods are predominantly used, with meteorologists often manually performing bias correction on some results. Our method offers a deep learning-based approach for more accurate correction in extreme precipitation forecasting.
>
> 2. Our benchmark can be used to develop efficient DL models as additions to NWP methods for precipitation forecasting. Given that this area remains under-explored, our benchmark and research have significant practical value and are poised to stimulate further studies in this domain.

---

> ### Author Response · Authors · 2023-11-22
> **Response to Reviewer hWvd-Part2**
>
> Q1.3: Experiments of different lead times.
>
> A1.3:
>
>  We have supplemented this in Section A.6 with additional results for lead times from 6 to 87 hours on the test set to provide a more detailed view of our model's performance over varying lead times.
>
> For the Korea dataset, we follow the setup of previous studies, selecting samples with various lead times ranging from 6 to 87 hours for our training, validation, and testing sets. The experimental results reported in our paper represent the average performance across different lead times within this 6-87 hour range.
>
> As shown in Figure 4 in the appendix of the revised PDF, within the lead time interval of 6 to 20, we observe that the CSI for rain reaches a peak at a lead time of 10 before exhibiting a declining trend, whereas the CSI for heavy rain peaks at a lead time of 9, subsequently showing a fluctuating trajectory. Expanding the analysis to a lead time range of 6 to 87, both rain and heavy rain CSI exhibit parallel trends, with heavy rain demonstrating superior performance over extended lead times, likely reflective of inherent data characteristics. Across all evaluated lead times from 6 to 87, our model's mean performance is enhanced, underscoring the comprehensive superiority of our modeling approach.
>
> For the Germany and China datasets, the lead time is set at a fixed 3 hours, corresponding to the paired format in which NWP predictions and precipitation observations are provided.
>
> In summary, while our approach adapts to the fixed 3-hour lead time in the Germany and China datasets, our comprehensive analysis and additional experiments on the Korea dataset underscore the versatility and robustness of our method across a range of lead times, affirming its applicability in varied precipitation forecasting contexts.
>
> Q1.4: Suggestion for reproducibility
>
> A1.4:
>
> We fully agree with the reviewer on the importance of reproducibility in benchmark studies, particularly in the field of climate science. To address this, we are committed to ensuring that our work is as transparent and replicable as possible.
>
> We have provided the code and data in the supplementary material.

---

> ### Author Response · Authors · 2023-11-23
> **Looking forward to your reply**
>
> Dear Reviewer  hWvd,
>
> Thank you very much again for the time and effort put into reviewing our paper. We believe that we have addressed all your concerns in our response. We have also followed your suggestion to improve our paper and have added additional experimental analysis. We kindly remind you that we are approaching the end of the discussion period. We would love to know if there is any further concern, additional experiments, suggestions, or feedback, as we hope to have a chance to reply before the discussion phase ends.
>
> Best regards,
>
> All authors

---

### Author Response · Authors · 2023-11-22
**Overall Comment**

We thank the reviewers for their useful suggestions. We sincerely thank all four reviewers for recognizing the value of our proposed benchmark and the usefulness of this application.

Major changes:

1. Add section  A.5 "Comparison with FourCastNet" in the appendix to add a sota method in weather forecasting as a baseline, which still shows our method's superiority. (Thanks for the suggestion from Reviewer #M7MM)
2. Add section  A.6 "Performance on different lead times on Korea Dataset" in the appendix to exploit our model's performance among different lead times. (Thanks for the suggestion from Reviewer #hWvd and #Lop7)
3. Add section  A.7 "Validation Loss on Germany Dataset" in the appendix to conduct qualitative analysis on the validation loss of Germany Dataset, which shows how the proposed channel attention and weighted loss work. (Thanks for the suggestion from Reviewer #hWvd)
4. Add our dataset link and code in the supplementary material. (Thanks for the suggestion from Reviewer #hWvd and #M7MM)

Overall strengths:

We thank the reviewers for recognizing the following strengths of our paper:

1. Valuable dataset for community: intriguing application of machine learning.(as noted by Reviewer #hWvd, #aiFF,  #Lop7)
2. All-round and detailed performance comparisons.(as noted by Reviewer #hWvd)
3. Noteworthy performance achievement: result significant.(as noted by Reviewer #M7MM, #Lop7 , #aiFF and #hWvd)

We reply to specific questions individually below.

---

### Meta-Review · Area_Chair_9vD4 · 2023-12-10

**Metareview:**

This paper introduces PostRainBench, a benchmark made from three Numerical Weather Processing-based (NWP) datasets (Korea, Germany, China) for heavy precipitation forecasting, and propose a multi-task (classification and regression) learning loss framework with a channel attention model; that model can outperform NWP on extreme precipitation.

Strengths:
* The paper is well written (hWvd)
* Significant performance improvement over NWP (hWvd,Lop7)
* Detailed performance comparisons with NWP (hWvd)

Weaknesses:
* Missing qualitative results to illustrated the impact of channel attention and weighted loss (hWvd): the authors added additional experiments
* Missing section on limitations (hWvd), addressed by the authors
* Unreleased code and data (hWvd): code and data are provided in the supplementary material
* No analysis for different lead times (hWyd,Lop7): in the rebuttal, the authors have provided predictions at horizons 3 to 87 hours.
* Limited technical contribution in the algorithm (Lop7,aiFF,M7MM) as it simply combines a UNet, the swinTransformer and a multi-objective loss
* Limited baselines and ablations (M7MM), partially adressed by the authors with new experiments.

Unfortunately, despite the engagement of reviewers in the discussion process, the authors did not succeed in convincing them and he final scores (3, 5, 6, 6) remain below the acceptance bar and I need to reject this paper.
Reviewer comments such as "intriguing application of machine learning, namely, precipitation prediction" (Lop7) do however suggest that this paper might find a better venue in an atmospheric science journal.

**Justification For Why Not Higher Score:**

The reviewers could not be convinced of the merit of the approach during the rebuttal phase.

**Justification For Why Not Lower Score:**

N/A

---

### Decision · Program_Chairs · 2024-01-16

Reject